# Numerical Simulation of Bubble and Velocity Distribution in a Furnace

Weitian Ding [1], Bing Qi [2], Huiting Chen [1], Ying Li [1], Yuandong Xiong [1], Henrik Saxén [3] and Yaowei Yu [1,*]

1 State Key Laboratory of Advanced Special Steel, Shanghai Key Laboratory of Advanced Ferrometallurgy, School of Materials Science and Engineering, Shanghai University, Shanghai 200444, China; dingweitian@shu.edu.cn (W.D.); huitingchen@shu.edu.cn (H.C.); yingli@shu.edu.cn (Y.L.); ydxiong@shu.edu.cn (Y.X.)
2 Quality Assurance Department, Shandong Iron&Steel Group Laiwu Branch, Jinan 271104, China; dijianyoulong@sina.com
3 Thermal and Flow Engineering Laboratory, Department of Chemical Engineering, Åbo Akademi University, Biskopsgatan 8, FI-20500 Åbo, Finland; henrik.saxen@abo.fi
* Correspondence: yaoweiyu@shu.edu.cn

**Abstract:** An industrial furnace, such as a blast furnace, molten salt furnace and a basic oxygen furnace, is a popular reactor, where the distribution of liquid, flow pattern of the fluid and the velocity of the fluid determine the energy distribution and chemical reaction in the reactor. Taking a furnace as the research object, this paper studies the effects of different inlet velocities, liquid densities and viscosity on bubble and velocity distribution. A three-dimensional mathematical model of the furnace is set up by a numerical simulation, and the volume-of-fluid (VOF) method is used to study the behavior of bubbles. The accuracy of the simulation parameters selected in the simulation calculation is verified by comparing the simulation with the experimental results. The findings show that an excessive or too small an inlet velocity will lead to an uneven distribution of chlorine in the furnace, therefore, an inlet velocity of about 30 m/s is more appropriate. In addition, changing the liquid density has little effect on the bubble and velocity distribution while choosing the appropriate liquid viscosity can ensure the proper gas holdup and fluidity of chlorine in the furnace.

**Keywords:** numerical simulation; bubble; VOF method

## 1. Introduction

Widely used in agriculture, military and the chemical industry, TiCl$_4$ is not only an important intermediate product for the production of titanium dioxide and sponge titanium, but also a solvent for dissolving and synthesizing organic substances such as plastics and resins [1–3]. The main methods of producing titanium tetrachloride in industry are molten salt chlorination and boiling chlorination [4–6]. In view of titanium ore containing CaO and MgO, molten salt chlorination is the best process choice [7,8].

Domestic and overseas scholars have undertaken a significant number of researches on the production of titanium tetrachloride. Hu [9] analyzed the main factors affecting molten salt chlorination, and reported a range of various factors in actual production. Ju et al. [10,11] established a finite element model to simulate the stable temperature field of a molten-salt chlorinator. The simulation results of the molten-salt chlorinator in stable production were similar to the field measured data. Morris et al. [12,13] studied the chlorination rate of titanium dioxide with CO and C as the reducing agents, and obtained the empirical equation of the ore reaction rate in a CO-Cl$_2$ system and C-Cl$_2$ system. Researches on the molten salt chlorination furnace have mainly focused on stable production, process control and optimization [14–17], and the chlorination reaction for the production of titanium tetrachloride largely concentrated in kinetics and thermodynamics [18–20]; however, most of the literature on the titanium slag chlorination furnace concerned research on its

production and technology, and much less is known about the distribution of chlorine after entering molten salt. To our knowledge, the distribution of chlorine in molten salt has an important impact on the subsequent chemical reaction, hence it is crucial to study the distribution of chlorine in a titanium slag chlorination furnace.

Since the process of chlorination is carried out at high temperature, the distribution inside the titanium slag chlorination furnace cannot be obtained experimentally, therefore, it is necessary to rely on numerical simulation methods. When the chlorine flow enters the molten salt from the bottom, it disperses into many small bubbles, involving bubble formation, breakup and coalescence [21–24]. At present, the numerical calculation methods of bubble behavior in a liquid based on the volume-of-fluid (VOF) model have been widely used. Akhtar et al. [25] studied the rising process of continuous bubble chains in a laboratory-scale bubble column by the VOF method. The results showed that small bubbles formed at lower superficial gas velocities and relatively larger bubbles were generated at higher gas velocities. Chen et al. [26] selected the VOF and *k-ε* turbulence model to simulate and analyze the distribution law of velocity, pressure and turbulent kinetic energy of the bubble generator along the flow direction. Gu et al. [27] numerically simulated the evolution of the wakes with different initial diameters that rise freely in still water via the VOF method. The results showed that the shape of the bubbles change from spherical to ellipsoidal during the rising. Wang et al. [28] used an improved VOF method to investigate the motion of a single bubble in ionic liquids. The simulation results of the deformation, velocity and equivalent diameter of the single bubble rising in the three ionic liquids were in good agreement with the experimental data, and the model was used to predict the detailed velocity and pressure fields around the bubbles.

It can be seen that the VOF model can well predict the movement of bubbles in a liquid and the gas–liquid interface. Therefore, the object of this paper is to study the process of bottom-blowing chlorine stirring molten salt based on the VOF model. The commercial CFD software, ANSYS Fluent, is used. The remaining sections of the paper are organized as follows: first, the combined CFD (computational fluid dynamics)-VOF model, the experiment and simulation conditions are described in Section 2; subsequently, compelling results are presented and discussed in Section 3; and finally, the conclusions are drawn in Section 4.

## 2. Model Theory

### 2.1. Water Model Experiment

Based on the similarity principle, a water model experiment was used to verify the results of the simulation, and the bottom-blowing method was used. The schematic diagram of the model used in the water model experiment is shown in Figure 1. The equipment was made of plexiglass, so that the state of the water stirred by the gas jet could be recorded with the help of shooting equipment during the experiment. The shooting equipment used was a high-speed camera with a resolution of 1920 × 1080, and a shooting frequency of 0.01 s. Since this research mainly discusses the fluid flow behavior in the gas–liquid system, the ratio of the inertial force of the bottom fluid to the gravity plays a decisive role. Therefore, the corrected Froude number ($Fr'$) was adopted as the similarity criterion. According to the corrected Froude number, we obtained:

$$Fr' = Fr_l' \tag{1}$$

$$\frac{u^2}{gd} \times \frac{\rho_g}{\rho_l - \rho_g} = \frac{u_l^2}{gd_l} \times \frac{\rho_{gl}}{\rho_{ll} - \rho_{gl}} \tag{2}$$

after the conversion:

$$\frac{Q_N}{Q_{Nl}} = \sqrt{\frac{d}{d_l} \times \frac{\rho_{gl}}{\rho_g} \times \frac{\rho_{ll} - \rho_{gl}}{\rho_l - \rho_g}} \tag{3}$$

where $d$ and $d_l$ are the diameter of the prototype and model, respectively (m); $u$ and $u_l$ denote the gas velocity of the prototype and model (m/s); $\rho_l$ and $\rho_{ll}$ represent the liquid density of the prototype and model (kg/m³); $\rho_g$ and $\rho_{gl}$ are the gas density of the prototype and model (kg/m³); $Q_N$ and $Q_{Nl}$ denote the gas flow of the prototype and model (m³/h); and $g$ is the acceleration of gravity, 9.81 m/s². The physical parameters of prototype and model are shown in Table 1.

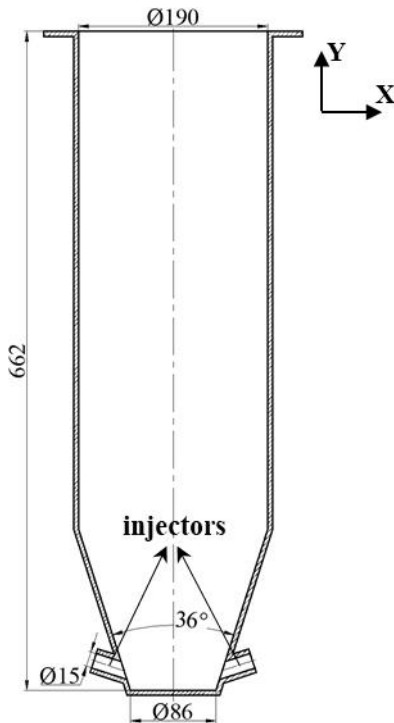

**Figure 1.** Schematic diagram of water model experiment (mm).

**Table 1.** Physical parameters of prototype and model.

| Parameters | Liquid/Gas | Density/kg/m³ | Viscosity/Pa·s |
|---|---|---|---|
| Prototype | Molten salt | 1650 | 0.0055 |
|  | Chlorine | 2.9 | $1.4473 \times 10^{-5}$ |
| Model | Water | 1000 | $1.01 \times 10^{-3}$ |
|  | Air | 1.29 | $1.79 \times 10^{-5}$ |

*2.2. CFD-VOF Model*

2.2.1. Volume of Fluid Model (VOF)

The VOF model can be used to track the phase interface. The tracking of the interface of the phases was accomplished by the solution of a continuity equation of the volume fraction of the phases. For one phase, this equation has the following form:

$$\frac{1}{\rho_q}[\frac{\partial}{\partial t}(\alpha_q \rho_q) + \nabla(\alpha_q \rho_q \vec{u_q})] = 0 \tag{4}$$

where $\alpha_q$ is the volume fraction of phase $q$, and $\rho_q$ is the density of phase $q$.

The primary phase volume fraction was obtained by the following constraint:

$$\sum_{q=1}^{n} \alpha_q = 1 \tag{5}$$

A single momentum equation was solved throughout the domain, and the obtained velocity field was shared among the phases. The momentum equation is dependent on the volume fractions of all phases through the properties $\rho$ and $\mu$ as follows:

$$\frac{\partial}{\partial t}(\rho \vec{u}) + \nabla \cdot (\rho \vec{u} \vec{u}) = -\nabla p + \nabla \cdot [\mu(\nabla \vec{u} + \nabla \vec{u}^T)] + \rho \vec{g} + \vec{F} \tag{6}$$

where $\rho = \sum \alpha_q \rho_q$, $\mu = \sum \alpha_q \mu_q$.

### 2.2.2. Turbulence Model (RNG $k$-$\varepsilon$ Model)

The bottom-blown gas–liquid two-phase flow is a complex mixing behavior, which produces a turbulence phenomenon with a high Reynolds number, showing a strong turbulent state with vortices, where the gas is sprayed into the molten salt at a certain speed, while the RNG $k$-$\varepsilon$ model has more advantages in dealing with the vortex flow with a large strain rate and streamline bending [29]. The RNG theory provides an analytically-derived differential formula for effective viscosity, which accounts for low-Reynolds number effects, hence the RNG $k$-$\varepsilon$ model is more accurate and reliable for a wider range of flow than the standard $k$-$\varepsilon$ model. Therefore, it was suitable to use the RNG $k$-$\varepsilon$ model to simulate bottom-blow stirring. The turbulence kinetic energy, $k$, and its rate of dissipation, $\varepsilon$, were obtained from the following transport equations:

$$\frac{\partial(\rho k)}{\partial t} + \frac{\partial(\rho k u_i)}{\partial x_i} = \frac{\partial}{\partial x_j}\left(\alpha_k \mu_{eff} \frac{\partial k}{\partial x_j}\right) + G_k + G_b - \rho \varepsilon \tag{7}$$

$$\frac{\partial(\rho \varepsilon)}{\partial t} + \frac{\partial(\rho \varepsilon u_i)}{\partial x_i} = \frac{\partial}{\partial x_j}\left(\alpha_\varepsilon \mu_{eff} \frac{\partial \varepsilon}{\partial x_j}\right) + C_{1\varepsilon}\frac{\varepsilon}{k}(G_k + C_{3\varepsilon}G_b) - C_{2\varepsilon}\rho\frac{\varepsilon^2}{k} \tag{8}$$

where $G_k$ represents the generation of turbulence kinetic energy due to the mean velocity gradients; and $G_b$ is the generation of turbulence kinetic energy due to buoyancy. $G_k$ and $G_b$ use the initial settings of the software. The quantities $\alpha_k$ and $\alpha_\varepsilon$ are the inverse effective Prandtl numbers for $k$ and $\varepsilon$, respectively. The correlated constants are given by theoretical analysis:

$$\mu_{eff} = \mu + \mu_t, \ \mu_t = \rho C_\mu \frac{k^2}{\varepsilon}, \ C_\mu = 0.0845, \ C_{1\varepsilon} = 1.42, \ C_{2\varepsilon} = 1.68.$$

where $C_{3\varepsilon}$ is calculated according to the following relation [30]:

$$C_{3\varepsilon} = \tanh\left|\frac{v}{u}\right|.$$

### 2.3. Geometric Model and Boundary Conditions

The schematic diagram of the geometric model used for the numerical simulation is shown in Figure 2. The titanium slag chlorination furnace consisted of a cylinder in the upper part and a cone in the lower part. Its total height was 8000 mm, the diameter of the cylindrical part was 3000 mm, and the diameter of the small end of the cone part was 1200 mm. In the cone section, four chlorine inlets with the diameter of 75 mm were evenly distributed at a height of 100 mm from the bottom, and the included angle between the inlet and the horizontal plane was 18°. The height of molten salt in the titanium slag chlorination furnace was 4000 mm.

This paper mainly studies the effects of different inlet velocities (15, 30, 60 m/s), different liquid densities (1000, 1261, 1650 kg/m$^3$) and different liquid viscosities (0.0055, 0.0100, 0.0500 Pa·s) on the bubble and velocity distribution in the furnace. The specific parameter settings are shown in Table 2. Hexahedral unstructured mesh was used for the numerical simulation and the mesh number was about 1.04 million. The mesh quality

refs to the rationality of mesh geometry and the mesh quality will affect the calculation accuracy. The mesh quality of this model was greater than 0.6.

The boundary condition of the chlorine inlet was the velocity inlet, the boundary condition of the furnace outlet was the pressure outlet, and the boundary condition of walls were a no-slip wall. The flow condition was an unsteady flow, and the pressure-velocity coupling was solved by the SIMPLEC algorithm. The momentum term adopted the second-order upwind and the turbulent term adopted the first-order upwind.

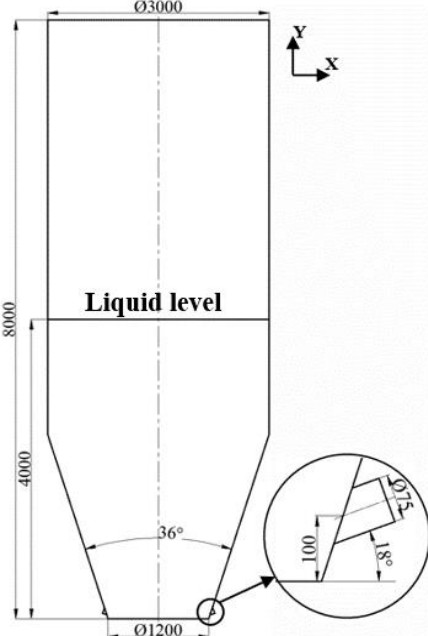

**Figure 2.** Schematic diagram of the geometric model (mm).

**Table 2.** The setting of simulation parameters.

| No. | Inlet Velocity/(m/s) | Liquid Density/(kg/m$^3$) | Liquid Viscosity/(Pa·s) |
|-----|------|------|------|
| 1 | 15 | | |
| 2 | 30 | 1650 | |
| 3 | 60 | | 0.0055 |
| 4 | | 1261 | |
| 5 | 30 | 1650 | |
| 6 | | 1650 | 0.0100 |
| 7 | | | 0.0500 |

## 3. Results and Discussion

### 3.1. Model Verification

Through the calculation of Equation 8, the gas flow used in the simulation and water model experiment were 0.627 m$^3$/h and 0.264 m$^3$/h, respectively. The captured pictures of the water model experiment were compared with the numerical simulation results to verify the reliability and the accuracy of the simulation. The experiment was to inject gas from the bottom inlet, and the research was carried out from the formation of the first bubble until the first bubble reached the liquid level. The time of the simulation results was 0.10 s, 1.20 s, 2.00 s and 3.10 s, respectively. The time of the experimental results was 0.01 s, 0.13 s, 0.25 s and 0.37 s. The bubble formation stage is provided in Figure 3a,a'. With the gradual increase in gas flow, the bubble continued to grow until it left the inlet. With the bubble rising, the bubble volume became larger due to the decreasing pressure of fluid as shown in Figure 3b. Because of the surface tension and buoyancy, the bubble presented a spherical-cap pattern as shown in Figure 3c. When the relative velocity was high enough to make the bubble skirt unstable, due to interfacial instability, the bubble skirt sheared off

from the spherical-cap bubble with the generation of small bubbles at the edge. The small bubbles continued to coalesce to form large bubbles, and finally broke up after reaching the liquid level as shown in Figure 3d.

By comparing the bubble motion, it can be found that the simulation was similar to the experimental results. This shows the accuracy of the simulation parameters selected in the simulation calculation and confirms the feasibility of establishing a mathematical model for the numerical simulation. This provides a guarantee for follow-up work.

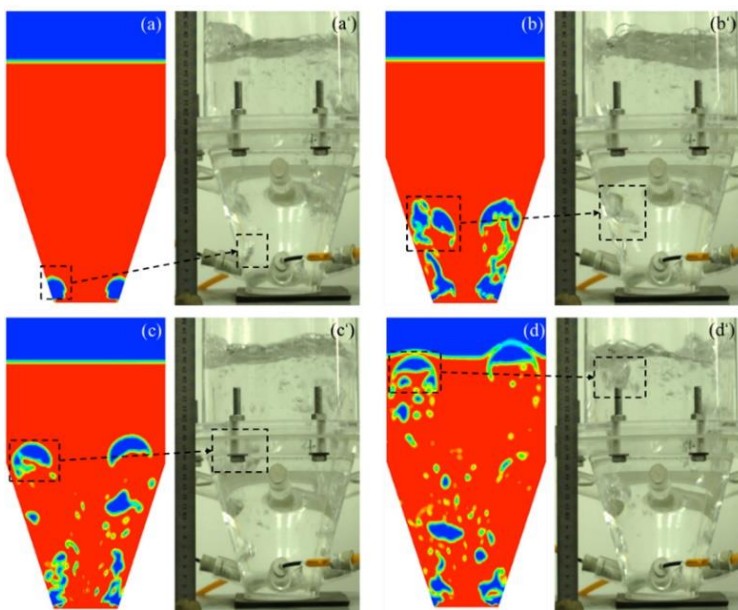

**Figure 3.** Comparison of simulation results and experimental results of the bubbles' rise (**left**: simulation model (**a**–**d**), **right**: water model (**a′**–**d′**)).

### 3.2. Distribution of Bubbles

Figures 4–6 show the distribution of bubbles at different inlet velocities (15, 30, 60 m/s), different liquid densities (1000, 1261, 1650 kg/m$^3$) and different liquid viscosities (0.0055, 0.0100, 0.0500 Pa·s). It can be seen from Figure 4 that, with the increase in inlet velocity, the depth of the chlorine entering molten salt and the number of bubbles increased. The depth with an inlet velocity of 60 m/s was about four times the depth with an inlet velocity of 15 m/s. When the inlet velocity was 15 m/s, the bubbles entered the furnace from the bottom inlet, moved upward along the wall, and broke at the liquid level, resulting in a small liquid level fluctuation. When the inlet velocity reached 60 m/s, the bubbles moved upward from the middle, leading to a severe liquid level fluctuation. Therefore, as the inlet velocity increased, the bubbles concentrated towards the middle, and the liquid level fluctuation became more intense. As shown in Figure 5, with the increase in liquid density, the number of bubbles was almost the same, the number of large bubbles decreased, and the fluctuation range of the liquid level was relatively small. Therefore, changing the liquid density had little effect on the distribution of bubbles in the furnace. As for Figure 6, with the liquid viscosity increase, the number of bubbles increased, and the fluctuation of the liquid level reduced. Therefore, controlling the liquid viscosity can ensure the gas holdup in molten salt for the chemical reaction.

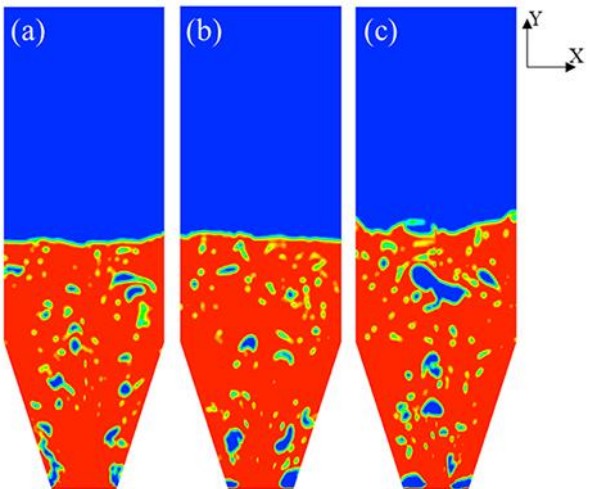

**Figure 4.** Distribution of bubbles at different inlet velocities; (**a**) 15 m/s, (**b**) 30 m/s, (**c**) 60 m/s.

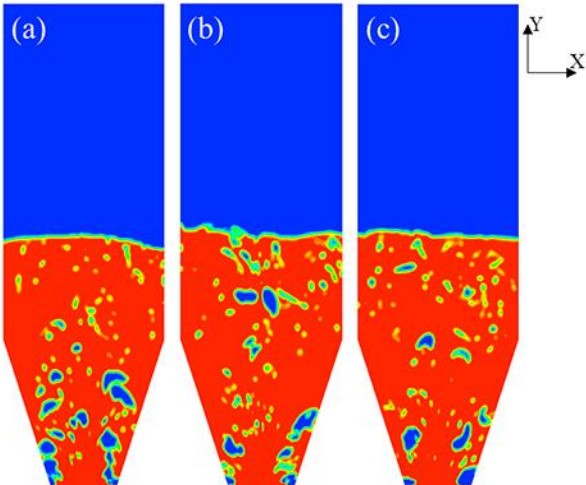

**Figure 5.** Distribution of bubbles at different liquid densities; (**a**) 1000 kg/m$^3$, (**b**) 1261 kg/m$^3$, (**c**) 1650 kg/m$^3$.

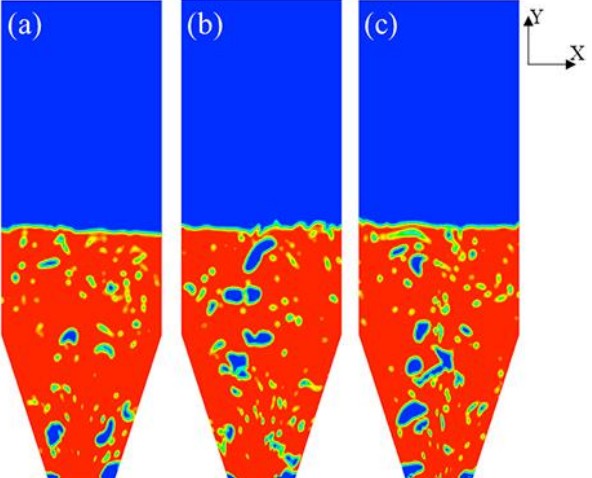

**Figure 6.** Distribution of bubbles at different liquid viscosities; (**a**) 0.0055 Pa·s, (**b**) 0.0100 Pa·s, (**c**) 0.0500 Pa·s.

### 3.3. Velocity Distribution

In order to better analyze the velocity distribution, the study was carried out near the inlet. Figures 7–9 show the velocity distribution in the *x*-axis direction under different parameters (inlet velocities, densities and viscosities), which was located in the center of the inlet and the center of the inlet was at *y* = 0.1 m from the furnace bottom. It can be seen that the trend of velocity distribution along the *x*-axis direction was roughly the same, and there was a low-speed area in the middle As shown in Figure 7, when the inlet velocity was 60 m/s, there was a small increase in velocity after the first decrease. This is because the inlet velocity was too large, which had a secondary impact on the interior, therefore, there was a second velocity increase and the excessive inlet velocity was concentrated chlorine in the middle, resulting in an uneven distribution of chlorine in the molten salt. The velocity distribution under different liquid densities shown in Figure 8 is roughly the same. In Figure 9, when the liquid viscosity was 0.0500 Pa·s, the speed of chlorine entering the molten salt decreased rapidly to the lowest. As shown in Figures 8 and 9, the liquid density and viscosity had little effect on the velocity distribution near the inlet.

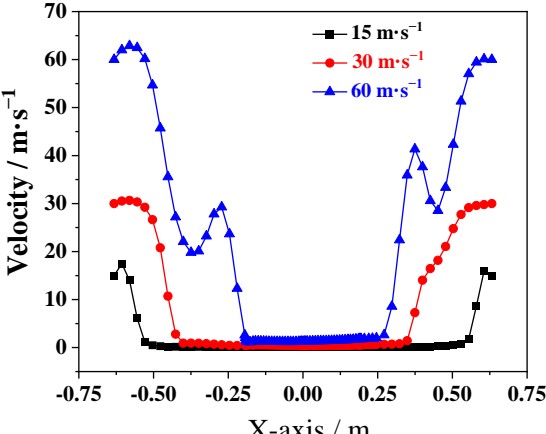

**Figure 7.** Velocity distribution in the *x*-axis direction under different inlet velocities (density: 1650 kg/m$^3$, viscosity: 0.0055 Pa·s).

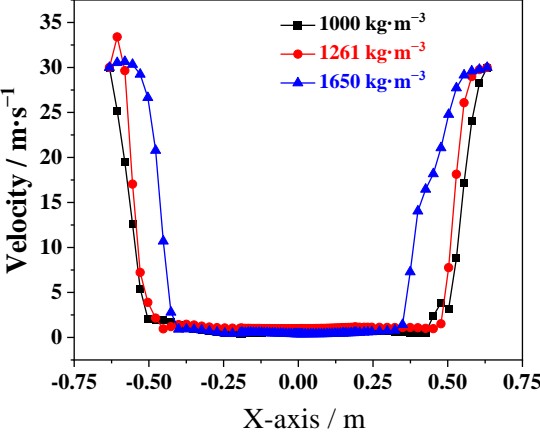

**Figure 8.** Velocity distribution in the *x*-axis direction under different liquid densities (inlet velocity: 30 m/s, viscosity: 0.0055 Pa·s).

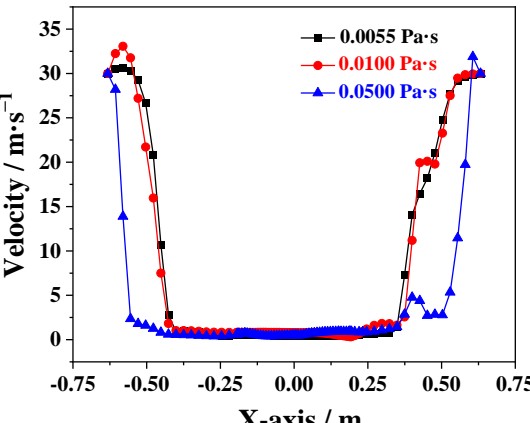

**Figure 9.** Velocity distribution in the *x*-axis direction under different liquid viscosities (inlet velocity: 30 m/s, density: 1650 kg/m$^3$).

### 3.4. Average Turbulent Kinetic Energy

The average turbulent kinetic energy under different conditions was analyzed. As shown in Figure 10, the average turbulent kinetic energy and its fluctuation amplitude increased with the increase in inlet velocity. When the inlet velocity was 60 m/s, the average turbulent kinetic energy fluctuated all the time, and the maximum peak value reached 2.72 m$^2$/s$^2$. The average turbulent kinetic energy with an inlet velocity of 15 m/s and 30 m/s showed an upward trend as a whole. This shows that the increase in inlet velocity was conducive to increasing the ability of gas–liquid mixing and stirring the molten salt. It can be seen from Figure 11 that the average turbulent kinetic energy under the different liquid densities showed an upward trend as a whole, and the oscillation amplitude in the first 2 s was large. The average turbulent kinetic energy basically reached a dynamic balance after 7.5 s, and the value fluctuated around 0.51 m$^2$/s$^2$. In Figure 12, the average turbulent kinetic energy under different fluid viscosities showed an upward trend before 5.2 s. Within 5.2 s to 7.2 s, the average turbulent kinetic energy fluctuated greatly around 0.59 m$^2$/s$^2$. The oscillation amplitude of the average turbulent kinetic energy after 7.2 s was small, and the value was about 0.55 m$^2$/s$^2$. This shows that after reaching a dynamic balance, the average turbulent kinetic energy under different liquid densities and viscosities was roughly the same.

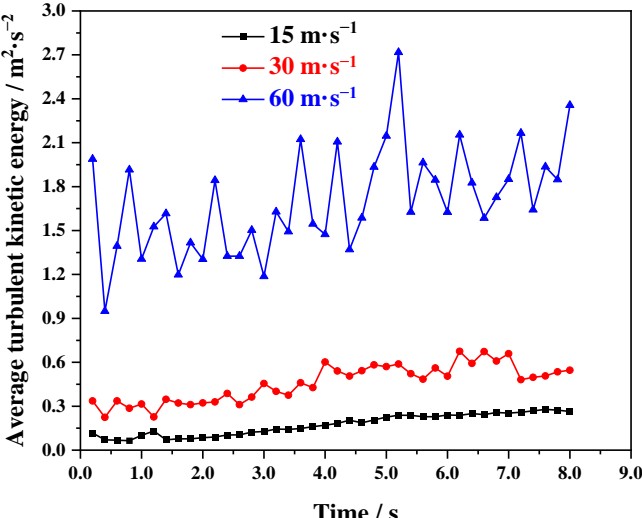

**Figure 10.** The average turbulent kinetic energy varies with time under different inlet velocities (density: 1650 kg/m$^3$, viscosity: 0.0055 Pa·s).

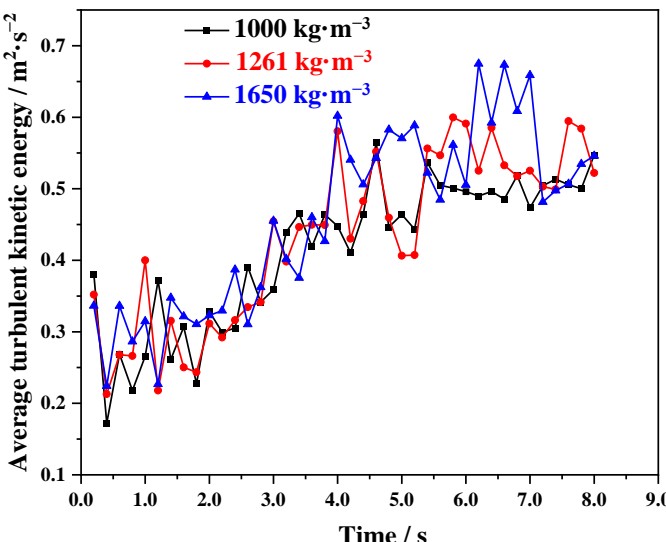

**Figure 11.** The average turbulent kinetic energy varies with time under different liquid densities (inlet velocity: 30 m/s, viscosity: 0.0055 Pa·s).

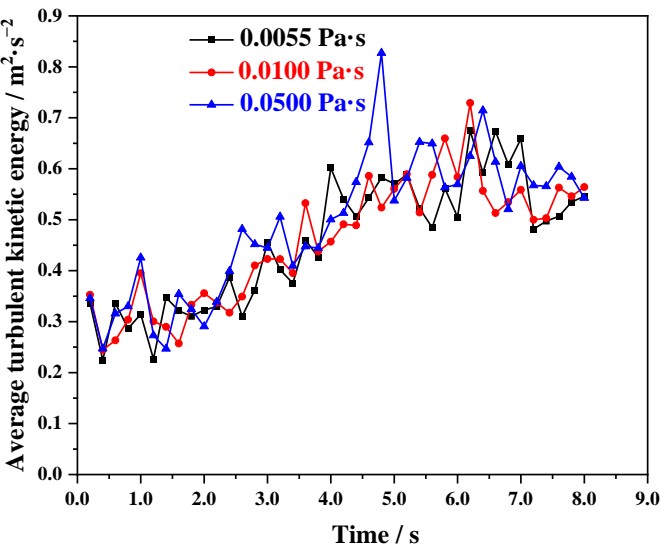

**Figure 12.** The average turbulent kinetic energy varies with time under different liquid viscosities (inlet velocity: 30 m/s, density: 1650 kg/m$^3$).

## 4. Conclusions

In this paper, the numerical simulation of titanium slag chlorination furnace was carried out with the simulation method. The effects of different inlet velocities, liquid densities and viscosities on the bubble and velocity distribution in titanium slag chlorination furnace were studied. The main conclusions are drawn as follows:

1.  Comparing the behavior of bubble motion, the simulation results were similar to the experimental results. This shows the accuracy of the simulation parameters selected in the simulation calculation and confirms the feasibility of establishing a mathematical model for numerical simulation.
2.  As the inlet velocity increased, the bubbles concentrated towards the middle, and the liquid level fluctuation became more intense. The depth of chlorine entering the molten salt with an inlet velocity of 60 m/s was about four times the depth with an inlet velocity of 15 m/s. In terms of velocity distribution, excessive or too small an inlet velocity may lead to the uneven distribution of chlorine in molten salt. Therefore,

an inlet velocity of about 30 m/s is more appropriate. To find a more accurate inlet velocity, further research is needed.

3.  With the increase in liquid density, the number of bubbles was almost the same, the fluctuation range of liquid level was relatively small, and the velocity distribution was very similar. Therefore, changing the liquid density had little effect on the bubble and velocity distribution.

4.  Although the increase in liquid viscosity increased the gas holdup, it resulted in the poor fluidity of bubbles in the molten salt. In order to ensure that the gas holdup and fluidity of chlorine in molten salt are proper, it is necessary to select the appropriate liquid viscosity to facilitate the subsequent chemical reaction.

5.  The average turbulent kinetic energy and its fluctuation amplitude increased with the increase in inlet velocity. This shows that increasing the inlet velocity is beneficial to enhancing the stirring effect. After reaching a dynamic balance, the average turbulent kinetic energy under different liquid densities and viscosities was roughly the same while the value of average turbulent kinetic energy under different liquid densities was $0.51\ \mathrm{m^2/s^2}$ and under different viscosities was $0.55\ \mathrm{m^2/s^2}$.

**Author Contributions:** Conceptualization, W.D.; methodology, W.D. and Y.Y.; software, W.D.; validation, W.D. and H.C.; formal analysis, Y.X.; investigation, B.Q.; resources, Y.Y.; data curation, W.D. and Y.L.; writing—original draft preparation, W.D.; writing—review and editing, H.S. and Y.Y.; visualization, Y.L.; supervision, W.D.; project administration, Y.Y.; funding acquisition, Y.Y. All authors have read and agreed to the published version of the manuscript.

**Funding:** This research received no external funding.

**Institutional Review Board Statement:** Not applicable.

**Informed Consent Statement:** Not applicable.

**Data Availability Statement:** The data presented in this study are available on request from the corresponding author.

**Conflicts of Interest:** The authors declare no conflict of interest.

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
