# Peer review of "Numerical Simulation of Bubble and Velocity Distribution in a Furnace"

_metals, doi:10.3390/met12050844_

Round 1

Reviewer 1 Report

In the abstract (not even in keywords) there is no mention of numerical simulation of titanium slag.

Indicate the meaning of the abbrevation CFD (row 72).

In eq. (1), (2) the meaning of α and ρ is not described.

Row 107 (in 2.2 Water Model Experiment).... the state of the molten pool?  In water  model experiment is molten pool?

Why you chose the selected values of velocities, liquid densities and liquid viscosities?

Fig. 3 add (water model), make image description more detaily - below the image.

Row 146....gas flow (air or chlorine?) 

Reviewer 2 Report

This papers presents the results of an investigation of the multiphase flow in a molten salt furnace that estimates the impact of various operating conditions (flow rate, density and viscosity) on the bubble and velocity distributions in the furnace, based on numerical simulations. The simulation method is first validated against a water model experiment.

While this paper is overall well written, beside a few typos, and the research it presents is easy to follow, the results that are presented are too superficial and not investigated deeply enough to support strong conclusions. The simulation work conducted by the authors seems of good quality, but the processing of their data and the discussion are insufficient in the current draft.

First, the details to describe the simulation method are not well chosen. It seems that the simulations were conducted using a commercial CFD software, but it is never indicated which one. The model theory shows a few equations, but these are insufficient to understand how the VOF method works. For example, the expression of properties of the multiphase mixture, ρ and µ in Eq. (3), is not discussed, so that Eq. (3) looks like a a regular single phase momentum equation. In Eq. (4) and (5) the terms Gk and Gb and defined but it is not written how they are set. The simulations are thus impossible to reproduce from what is described in the paper. The expression for the term C is not easy to understand, at least it is given, but it should also be supported by a reference justifying this choice. The “mesh quality” (line 136) is not defined, so a value of 0.6 does not mean much.

The same lack of information appears in the water model description. For example, the “shooting equipment” is not described at all (device, resolution, acquisition frequency, etc). The viscosity in the model is not given, and not accounted for in the Froude number, which becomes an issue since the effect of viscosity in investigated later and seems to play a role in the operation of such a furnace.

Figure 3 is very hard to read and is not very convincing as validation. First, zooms of the regions of interest should be provided for a better visual comparison between simulations and experiments to be possible. Second, an only visual comparison is not very strong. Third, the times of the different snapshots are not indicated so that the comparison does not make much sense. At least, the legend should indicate the snapshot times of each subfigure (a, b, c, d) in both simulation time and model time, the correspondence of which can be calculated and must be detailed in the model scaling along with Eq. (6) to (8).

The parametric study of the operation is not well described. While the varying parameters are given, the constant values are not. The values should be provided for ρ in Fig. 4 and 6, µ in Fig. 4 and 5 and Q in Fig. 5 and 6, maybe in the form a table summarizing all the investigated operating conditions. Moreover, the evaluation criteria to estimate the impact of theses parameters are not clear: for example, what is the meaning of the “fluidity of bubbles in the liquid” (line 181) and how is it estimated? Regarding the results themselves, everything is presented in the form of snapshots that, by definition, carry no statistical information. Comparing the instantaneous volume fraction maps is thus not very meaningful, it would not be surprising that the same operating conditions could produce all subpictures a, b and c of all Fig. 4, 5 and 6 at different times. At least, the snapshot time should be given. Now, some differences in bubble shapes can be observed, which my justify using a few snapshots to characterize bubble distribution, but it makes no sense to use the same approach for velocity fields. Average and standard deviation (fluctuation) velocity fields over long operating times would provide much more relevant information. To evaluate the mixing efficiency in the furnace, those values would not even be enough, since the simulations rely on a modelled turbulence, so turbulence maps (of k or ε) would be necessary to conclude about the “uneven distribution of chlorine” (line 229). Or better yet, a simulation with scalar transport would even provide quantitative information. In the current draft, only qualitative observations are drawn while simulations provide much more information. Fig. 7 to 9 are very hard to read. Arrows are very hard to compare, and the colorscales are useless: the scales differ between graphs, and even on single graphs the red regions are too local so that all the flow appears in pale variations of blue that cannot be quantitatively interpreted. Even the qualitative observations are not clear: key sentences in the discussion are worded in a very unclear way. The “swirling zones” (line 197) could be seen by the reviewer, and the whole paragraph between lines 225 and 230 is very confusing, while also being the only place where the impact of velocity inlet on operation is discussed in therms of fundamental mechanisms at play.

It is not indicated where the profiles from Fig. 10 to 12 have been taken, what is their position on the y axis? It is also not indicated if these are averaged (how?) or instantaneous profiles.

In the end, the papers shows a lot of good looking pictures but does not extract much information from the simulations, and consequently only draw qualitative conclusions that are only weakly supported by the simulation results. The simulations seem to be of good quality but have not been sufficiently processed, analysed and discussed in the current draft to be of scientific interest.

Regarding the form of the paper, there also some issues. It does not make much sense to put section 2.2 on the water model, between the simulation method and the geometry of the simulation domain.

Plus, the bibliography is completely broken. It seems that the first and last names of the authors in most (all ?) references are inverted, if even present (see Ref. [12] for example). The citations do not even appear with the same author list in the reference list as where they are cited. The citations are weird too: when summarizing the authors with et al., only the first author name is necessary. Also, this draft lacks comparison with the scientific literature. All citations appear before section 3 (actually even before section 2.2) meaning that the results are never discussed in the light of other studies, which severely limits the scope of this investigation and reflects a much too superficial analysis of the results.

Round 2

Reviewer 2 Report

Although the reviewer still gets the impression that could benefit from a deeper analysis of the simulation results, the authors have addressed the important points and have made the required changes to their first draft to turn it into a satisfactory article.
The numbers in the first column in Table 2 should be referenced to in the legends of Figures 10 to 12 to make it easier to understand which operating condition sets are compared with one another.